# A meta-analysis of the value of circulating tumor cells in monitoring postoperative recurrence and metastasis of colorectal cancer

Jiao Wu[1][☉], Zhongyu Li[1][☉], Jianhua Zou[1][‡], Liusheng Li[1][‡], Ning Cui[1], Tengteng Hao[1], Kangjun Yi[2], Jingyan Yang[2], Yu Wu[1,2]*

1 Oncology Department of Xiyuan Hospital, China Academy of Chinese Medical Science Haidian District, Beijing, China, 2 Graduate School of Beijing University of Chinese Medicine, Beijing, China

☉ These authors contributed equally to this work.
‡ JZ and LL also contributed equally to this work.
* zoujianhua2021@163.com

**Data Availability Statement:** All relevant data are within the paper and its Supporting Information files.

## Abstract

### Objective

Circulating tumor cells (CTCs) as novel biomarkers are widely investigated in various cancers, although most of the literature shows that CTCs have predictive value for recurrence, metastasis, and prognosis after CRC surgery, results remain controversial. We aimed to systematically evaluate the value of CTCs in monitoring of colorectal cancer (CRC) recurrence and metastasis after surgery.

### Method

The PubMed, Cochrane Library, Embase, and other databases were searched from the establishment of the database to May 27, 2021. Relevant literature searches and data extraction were performed independently by two reviewers. The quality assessment was performed using the QUADAS2 scale developed by the Cochrane collaboration. The heterogeneity was checked using the Spearman correlation coefficient and the Cochran-Q test in the Meta-Disc1.4 software. Subgroup analysis was used to explore the source of heterogeneity. Considering that all the included papers were clinical studies with clinical heterogeneity, random effect model was adopted for analysis. And the sensitivity (Sen), specificity (Spe), positive likelihood ratio (PLR), negative likelihood ratio (NLR), diagnostic odds ratio (DOR), and summary receiver operating characteristic (SROC) curves of CTCs, in monitoring recurrence and metastasis after CRC, were calculated. The publication bias of the included studies was assessed using Deek's funnel figure.

### Result

The literature included a total of 13 articles, comprising 1788 cases, and the overall quality of the literature was high. After summing up the indicators, the sensitivity pooled-value of

**Funding:** This study was supported by grants from the National Key Research and Development Plan of China (2017YFC1700606) and Major Key Projects of Science and Technology Innovation Project of China Academy of Chinese Medical Sciences (CI2021A01803). The Funder is the corresponding author of this manuscript and helped design it.

**Competing interests:** The authors have declared that no competing interests exist.

the peripheral blood CTCs to monitor the recurrence and metastasis value of CRC after CRC was 0.67 [95%*CI* (0.62, 0.71)], specificity pooled-value was 0.71 [95%*CI* (0.67, 0.72)], PLR pooled-value was 2.37 [95%*CI* [1.52, 3.71]), NLR pooled into 0.53 [95%*CI* (0.36, 0.78)], DOR pooled into 4.97 [95%*CI* (2.11, 11.72)], AUC was 0.7395.

## Conclusion

Peripheral blood CTCs have a moderate monitoring value for recurrence and metastasis after CRC; CTCs detected one week after surgery may be more correlated with recurrence and metastasis.

## Introduction

Colorectal cancer (CRC) is one of the diseases with high morbidity and mortality worldwide [1], In the incidence of malignant tumors in China, male incidence ranks third and mortality ranks fifth. Female morbidity and mortality ranked third [2]. Surgery is still the preferred treatment for patients with early CRC [3], However, the postoperative recurrence and metastasis rate is high, nearly half of CRC patients have recurrence after radical surgery [4], and a significant number of patients still die of tumor recurrence and metastasis after radical surgery [5]. The 5-year survival rate was significantly reduced due to recurrence and metastasis after CRC [6], Postoperative recurrence and metastasis is one of the main reasons for the low quality of life and survival rate of CRC patients. Therefore, monitoring the recurrence and metastasis of CRC after surgery and selecting the appropriate treatment plan can help improve the quality of life of patients and prolong their survival.

At present, NCCN guidelines recommend that carcinoembryonic antigen (CEA) and conventional imaging examinations (Ultrasonography, enhanced MRI, enhanced CT, etc.) should be used to monitor recurrence and metastasis in patients with CRC. However, the sensitivity and specificity of CEA in postoperative monitoring of recurrence and metastasis of CRC are not high [7], its use is limited. Due to fibrous hyperplasia and anatomical changes in the operative area after surgery, it is difficult to identify some recurrence and early metastasis in the abdominal and pelvic areas by conventional imaging [8]. Although the sensitivity and specificity of PET-CT in monitoring postoperative recurrence and metastasis of CRC are relatively high [9], PET/CT examination is expensive and has high requirements for imaging physicians, which affects its clinical promotion and use. Circulating tumor cells (CTCs) are tumor cells shed from primary or secondary tumor masses that circulate in the bloodstream [10], In the past, CTCs was often used in the early diagnosis, monitoring of recurrence and metastasis, prognosis and efficacy evaluation of malignant tumors such as lung cancer [11], breast cancer [12], prostate cancer [13] and colorectal cancer [14] and so on. However, there has been no unified view of the value of CTCs in postoperative recurrence and metastasis of CRC. There are still controversies regarding the monitoring of postoperative recurrence and metastasis of CRC. Although most literature has affirmed the value of CTCs in monitoring postoperative recurrence, metastasis and prognosis of CRC, some literature still didn't agree with this view [15]. In this study, we explored the value of CTCs in monitoring post-operative recurrence and metastasis of CRC patients by including relevant literature and extracting relevant data for meta-analysis, and we hope that this paper can provide clinical evidence for further clinical monitoring of CRC recurrence and metastasis.

## Materials and methods

This review is reported according to the PRISMA guidelines. The protocol was registered with the PROSPERO database (registration ID: CRD42021264495), which is an international perspective registry for systematic reviews.

### Literature search

We searched the PubMed, Cochrane Library, Embase, and other databases by computer. The search time was from the establishment of the database to May 27, 2021, and the language was limited to English. Using the combination of subject words and free words, the search terms included "Colorectal Neoplasm*", "Colorectal Cancer*", "Colorectal Tumor*", and "Circulating Tumor Cell*", "Recurrence*, Recrudescence*", "Relapse*", "Metastatic, Neoplasm", "Metastases", "metastatic, Neoplasm", etc. In order to avoid omission, reference lists of relevant literature were also browsed, and articles meeting the inclusion criteria were included and screened for full text reading. Literature screening was conducted by two researchers (YJY and YKJ).

### Inclusion criteria and exclusion criteria

Papers meeting the following criteria were included: 1) Prospective and retrospective cohort studies; 2) The main subjects were confirmed to be colorectal cancer by pathological examination and underwent surgical treatment, with TNM stage I-III; 3) Peripheral blood CTCs data were all detected after surgery, and CTCs detection methods were clearly described; 4) Outcome indicators: The results included in the literature should contain or be calculated to obtain the following data: true positive value (TP), false positive value (FP), false negative value (FN), and true negative value (TN).

Excluding literature with the following conditions: 1) Patients with a family history of colorectal cancer, patients without surgery for the primary site of colorectal cancer, and stage IV patients; 2) Animal experiments, cell experiments, reviews, systematic evaluation and meta-analysis, case reports, conference literature, etc.; 3) Insufficient data and repeated literature; 4) Blood samples taken from the bone marrow, mesenteric or portal vein, lymph nodes, and abdominal cavity, those with only pre-operative CTC data were excluded.

### Data extraction

Data was extracted independently by two reviewers (LSL and HTT) and then cross-checked by one another. The extracted data included the following: first author, publication year, country, number of cases, age, CTC detection method and sample collection time, adjuvant treatment or not, follow-up time, TP, FP, FN, and TN. The extracted data were placed in an Excel table.

### Quality evaluation

The quality assessment was performed using the QUADAS2 scale developed by the Cochrane collaboration [16]. Quality evaluation was independently evaluated by two researchers (ZJH and WJ) according to a unified standard. When there were disagreements, a third professional discusses his/her opinion and an agreement was reached.

### Statistical analysis

Heterogeneity test: The heterogeneity caused by the threshold effect was checked using the Spearman correlation coefficient in the Meta-Disc1.4 software. If $P > 0.1$, it indicates that there is no heterogeneity caused by the threshold effect; when $P \leq 0.1$, it indicates that there is a heterogeneity caused by the threshold effect. Meta-Disc1.4 software was used to draw the DOR

forest figure, and the Cochran-Q test was used to explore the heterogeneity caused by the non-threshold effects. According to the Cochrane System Intervention Evaluation Manual: when $I^2$ >50% or $P<0.1$, it indicates the existence of heterogeneity caused by the non-threshold effects. Subgroup analysis is used to explore the source of heterogeneity. Considering that all the included articles were clinical studies with clinical heterogeneity, random effect model was adopted for analysis [17]. Publication bias was evaluated Deek's funnel figure using Stata version 15.0.

Summary of analysis indicators: Meta-Disc 1.4 software was used to combine sensitivity, specificity, positive likelihood ratio (PLR), negative likelihood ratio (NLR), and diagnostic odds ratio (DOR); draw a summary receiver operating characteristics (SROC) curve; and used the Q value to represent the analysis result of SROC. The larger the Q value, the more accurate the diagnostic test [18]. The AUC of the area under the SROC curve was calculated. Assuming that the test level is $\alpha = 0.05$, the diagnostic value is judged as follows: 1) The diagnostic value is low: AUC = 0.5~0.7; 2) The diagnostic value is moderate: AUC = 0.7~0.9; and 3) The diagnostic value is high: AUC = 0.9~1.

## Results

### Literature search results

By searching the databases, we retrieved 1505 articles from PubMed, 516 from Embase, and 100 from Cochrane library; a total of 2121 articles were retrieved. Using Endnote document management, 319 duplicate articles and 1763 that did not meet the inclusion criteria were deleted. After reading the titles and abstracts, 39 articles remained. After reading the full text, 13 articles were finally included. Of the excluded articles, 7 contained stage IV cases; 6 of the excluded articles had no recurrence and metastasis data; 5 could not extract data; and 8 only had abstracts without full text (the retrieval process is shown in Fig 1).

### Basic characteristics of the included studies

The 13 articles included, of which 6 were from China [24–30]; 1 from Italy [28]; 1 from Poland [19]; 2 from Japan [20, 23]; 1 from Spain [19]; 1 from the UK [20]; and 1 from Croatia [15], comprised a total of 1788 cases. Ten were prospective cohort studies [15, 19, 21–25, 28–30], and three were retrospective cohort studies [20, 26, 27]. Regarding the detection methods of CTCs, one study used CellSearch detection and analysis [28], two papers used immunofluorescence in Situ Hybridization(imFISH) detection and analysis [29, 30], 6 papers used RT-PCR methods [15, 19–23], and four papers used membrane-array detection methods [24–27]; CTCs detection was performed in colorectal cancer patients 7 days after surgery in 5 studies [23–27], CTCs were detected within 7 days after surgery in 8 studies [15, 19–22, 28–30] (Table 1).The included literature was evaluated using the QUADAS2 quality evaluation scale, as shown in Fig 2. The overall quality of the study was high.

### Meta analysis results

Heterogeneity test: By using Meta-Disc1.4 software to draw the SROC curve (Fig 3A), it was found that there is no "shoulder-arm-like" distribution between the studies, and the spearman correlation coefficient between the sensitivity log and the (1-specificity) log is 0.115, $P = 0.707$, indicating that there is no threshold effect among the included studies (Fig 3B). The Cochran-Q test was used to analyze the heterogeneity caused by non-threshold effects, and the results showed that Cochran-q = 106.83, $P = 0.00$, and $I^2 = 88.8\%$(great heterogeneity) (as shown in Fig 3C), indicating that heterogeneity caused by non-threshold effects existed among the included studies.

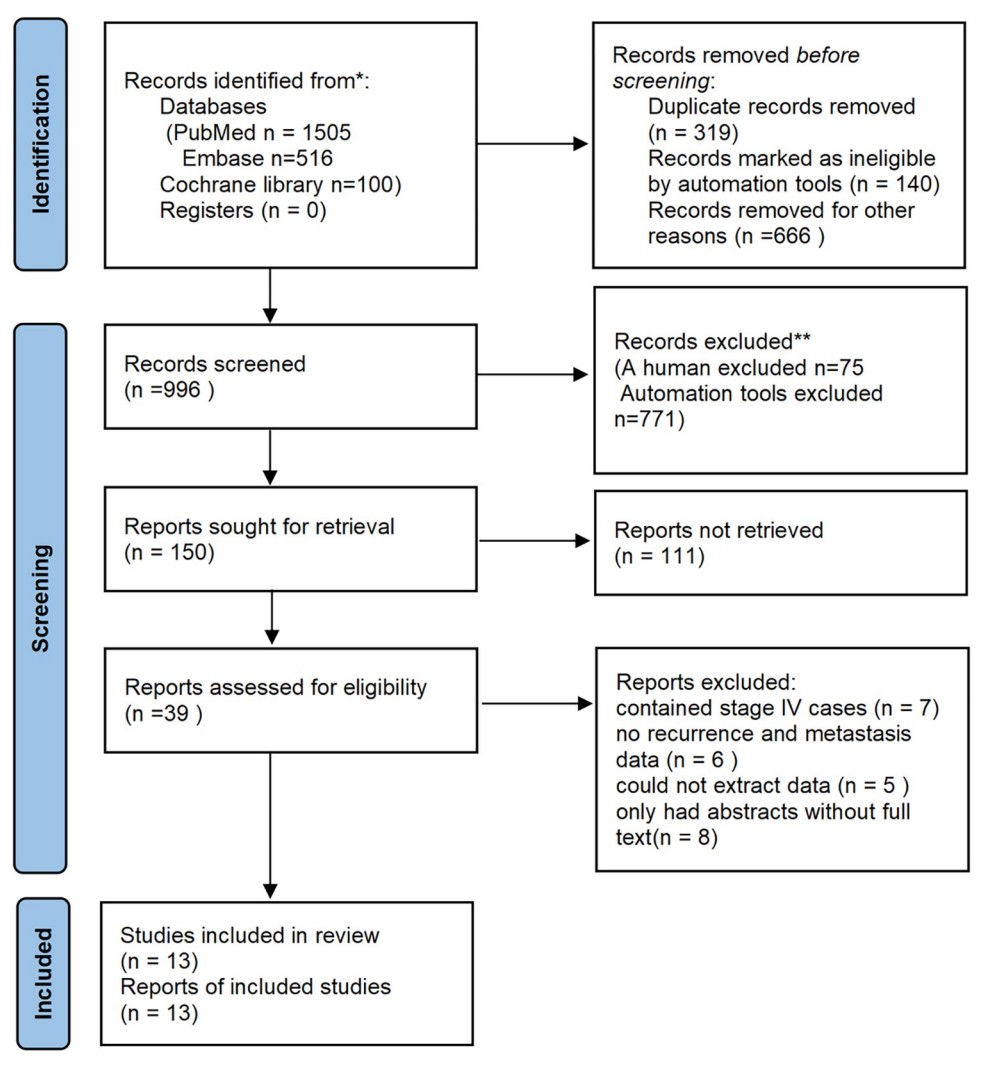

**Fig 1. Search flowchart.**

**Sensitivity analysis.** Stata software (version 15.0) was used to check the included literature one by one, and the results showed that every time a study was eliminated, the other studies were within the credible interval, and the effect on the combined effect size was small, indicating that the included studies were between the small difference indicates that the research results are more robust and the results of the analysis are more reliable (Fig 4).

**Combined analysis results.** Because the heterogeneity between the various studies is caused by non-threshold effects, the random effects model is used for the combination of statistics. The meta-analysis showed that the value of sensitivity pooled with CTCs in peripheral blood to monitor the recurrence and metastasis of colorectal cancer after surgery was 0.67 [95%*CI* (0.62, 0.71) (Fig 5A), the pooled specialty was 0.71 [95%*CI* (0.67, 0.72)] (Fig 5B), PLR pooled into 2.37 [95%*CI* (1.52, 3.71)] (Fig 6A); NLR pooled into 0.53 [95%*CI* (0.36, 0.78)] (Fig 6B), DOR pooled into 4.97 [95%*CI* (2.11, 11.72)] (Fig 3C), AUC of 0.7395 (Fig 3A).

**Subgroup analysis.** The results of subgroup analysis showed that the sensitivity, specificity, DOR and AUC of the membrane array method were higher than those of RT-PCR. The

**Table 1. Characters of included studies.**

| author and publish year | country | N | average age | Tumor stage | CTCs detection method | threshold | Median follow-up time (months) | research methods | Postoperative adjuvant treatment | TP | FP | FN | TN | Collection time | Mark |
|---|---|---|---|---|---|---|---|---|---|---|---|---|---|---|---|
| Nesteruk2014 [21] | Poland | 91 | 66±11.5 | I-III | RT-PCR | 1/2.5ml | 36 | prospective | no | 7 | 38 | 5 | 41 | 7 days after resection | CEA/CK20/CD133 |
| Ito2002 [22] | Japan | 99 | NR | I-III | RT-PCR | 2 colon carcinoma cells in the range from 10 to $10^5$ cells per $1*10^7$ peripheral blood leukocytes | NR | retrospective | no | 6 | 20 | 6 | 67 | after surgery | CEA mRNA |
| Bessa 2003 [19] | Spain | 66 | ≤73 years: 35; >73 years: 31 | I-III | RT-PCR | 5/10 mL | 36 | prospective | yes | 8 | 28 | 7 | 23 | 24 hours after surgery | CEA mRNA |
| Allen-Mersh 2007 [20] | UK | 147 | 67.4±13.2 | Dukes' A-Dukes' C | RT-PCR | 2/7ml | 46 | prospective | part yes | 12 | 91 | 11 | 23 | 24 h after surgery | CEA or CK20 |
| Sadahiro2007 [23] | Japan | 200 | NR | I-III | RT-PCR | one CEA mRNA-expressing cancer cell in $1*10^5$ normal lymphocytes. | 52 | prospective | no | 20 | 24 | 35 | 121 | between 7 and 10 days after surgery | CEA mRNA |
| Yih-Huei2007 [24] | Chinese Taipei | 194 | 64.9(28–90) | II | Membrane-Arrays | 5/1ml | 40 | prospective | part yes | 45 | 8 | 11 | 130 | at least 1 week after surgery | hTERT; CK-19; CK-20; CEA |
| Yih-Huei2008 [25] | Chinese Taipei | 438 | 65.6±13.1 | I-III | Membrane-Arrays | 5/1ml | 41 | prospective | part yes | 88 | 49 | 42 | 259 | 1 week after operation | hTERT; CK-19; CK-20; CEA |
| Lu2011 [26] | Chinese Taipei | 141 | 64.1(30–88) | II-III | Membrane-Arrays | 5/1ml | 40 | retrospective | part yes | 37 | 14 | 11 | 79 | 4 weeks after operation | hTERT; CK-19; CK-20; CEA |
| Lu2013 [27] | Chinese Taipei | 90 | 63.1(32–80) | III | Membrane-Arrays | 5/1ml | 36 | retrospective | yes | 19 | 2 | 11 | 58 | 1 week and 4 weeks after completion of adjuvant chemotherapy | hTERT; CK-19; CK-20; CEA |
| Gazzaniga2013 [28] | Italy | 37 | NR | II-III | CellSearch | 1/7.5ml | 8 | prospective | NR | 1 | 7 | 0 | 29 | after resection | NR |
| Kust2016 [15] | Croatia | 82 | 66±9.6 | I-III | RT-PCR | 1/1ml | 50 | Prospective | NR | 22 | 39 | 4 | 17 | 5–7 days after tumor resection | CK20 |

(*Continued*)

**Table 1.** (Continued)

| author and publish year | country | N | average age | Tumor stage | CTCs detection method | threshold | Median follow-up time (months) | research methods | Postoperative adjuvant treatment | TP | FP | FN | TN | Collection time | Mark |
|---|---|---|---|---|---|---|---|---|---|---|---|---|---|---|---|
| Wang2019 [29] | China | 130 | 63 | II-III | imFISH | 2/3.2ml | follow-up of tumor recurrence was performed | Prospective | no | 15 | 48 | 3 | 54 | within three days after operation | |
| Yu2020 [30] | China | 73 | NR | II | imFISH | 2/7.5ml | 22.1 | Prospective | part yes | 14 | 34 | 2 | 23 | the 7th postoperative day | anti-CD45 |

NR: It's not stated in the article. imFISH: immunofluorescence in Situ Hybridization.

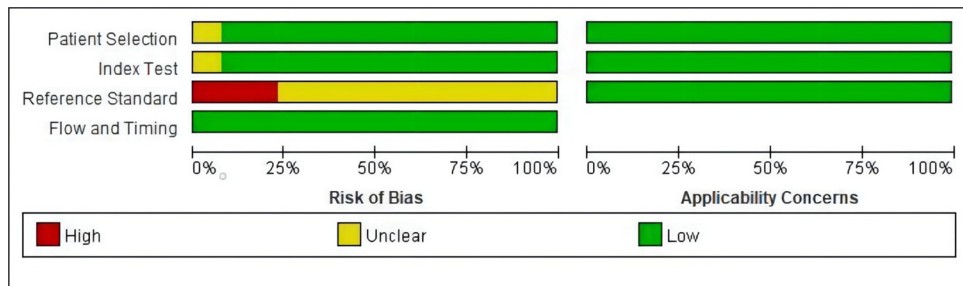

**Fig 2. Quality evaluation.**

sensitivity, specificity, DOR, and AUC values of CTCs detected one week after surgery were higher than those detected within one week after surgery. The results are shown in Table 2.

**Publication bias.** The Deeks symmetry test was performed with stata15.0 software, and the results showed that the funnel chart was basically symmetrical ($P = 0.23$), indicating that the included studies had no obvious publication bias (Fig 7).

## Discussion

CRC is a common clinical malignant tumor with a high postoperative recurrence and metastasis rate. About 25–50% of stage II-III CRC patients will have recurrence and metastasis after comprehensive treatment [31], in addition, the survival time after recurrence and metastasis is significantly shortened, and postoperative recurrence and high metastasis rate are important factors affecting the outcome of CRC [32]. 3 years after operation is the risk period for recurrence and metastasis of CRC [33]. Studies have shown that ctDNA and CTC are associated with pathogens of colorectal cancer. The most recent NCCN guidelines include this discussion regarding ctDNA: a prospective, multicenter study of 130 patients with stage I–III colon cancer detected ctDNA by multiplex, PCR-based next-generation sequencing (NGS). Thirty days after surgery, patients with positive ctDNA assays were seven times more likely to experience disease relapse than patients who were ctDNA negative (HR, 7.2; 95% CI, 2.7–19.0; P < .001). Likewise, after adjuvant chemotherapy, patients with ctDNA-positive assays were 17 times more likely to have disease relapse (HR, 17.5; 95% CI, 5.4–56.5; P < .001). Another prospective study of 150 patients with localized colon cancer detected dtDNA with NGS following surgery. In this study, detection of ctDNA was also associated with poorer DFS (HR, 17.56). However, ctDNA is more used in metastatic colorectal cancer [34, 35]. Some literature used ctDNA to monitor disease progression in colorectal cancer after early and medium-term, although the specificity was high (93%), its sensitivity was only 27% [36]. Circulating tumor cells (CTCs) are rare cancer cells that shed from tumor cells and remain in the circulating bloodstream, and are thought to be the "seeds" that initiate cancer progression and metastasis [14], At present, a number of clinical studies have confirmed that the existence of CTCs is closely related to recurrence and metastasis. Uen et al. [25] believed that early recurrence after stage I-III CRC was closely related to the persistent presence of CTC in peripheral blood. Lu et al. [27] performed CTCs detection on 141 patients with stage II and III colon cancer after surgery, and found that 51 of the 141 patients had persistent CTCs, and 37 of the 51 patients had early recurrence, suggesting that the persistent presence of CTCs in peripheral blood is a predictor of early recurrence in stage II and III colon cancer patients after surgery. Wang et al. [29] found that the CTC-positive patients with stage II and stage III CRC after surgery have a high risk of recurrence, and the recurrence-free survival rate of CTC-positive patients was significantly reduced.

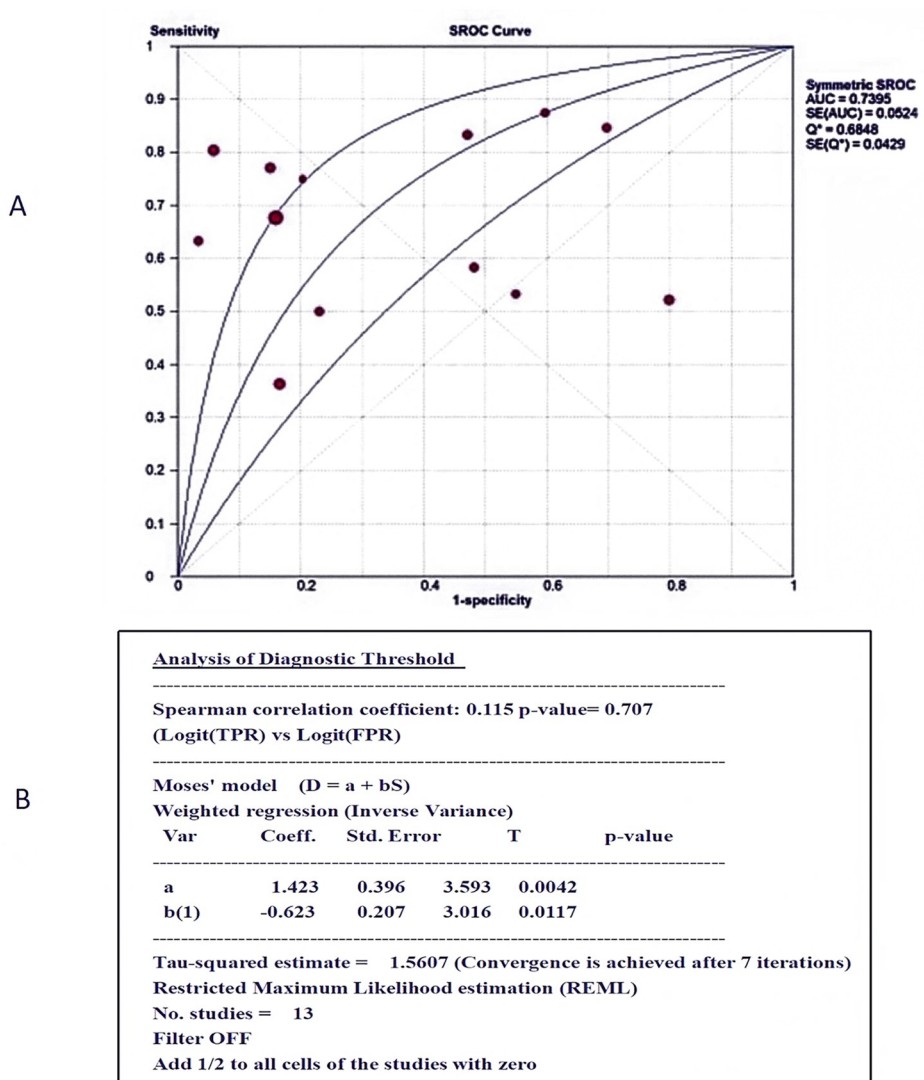

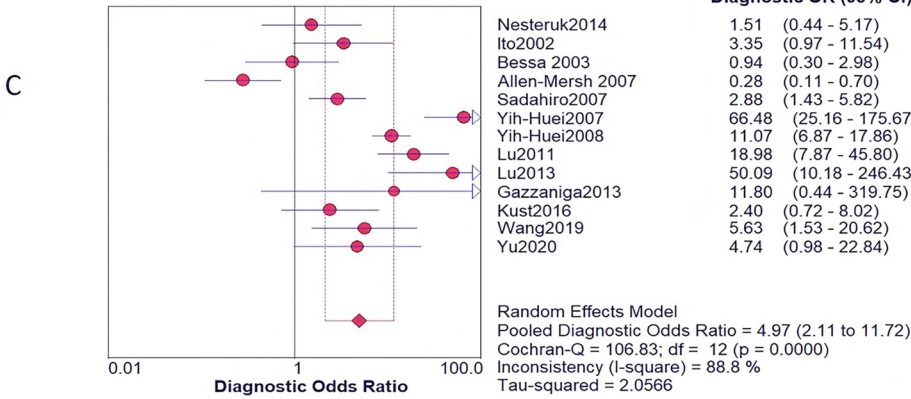

**Fig 3.** A: SROC curve. B: Analysis of threshold effect. C: Diagnostic odds ratio forest plot.

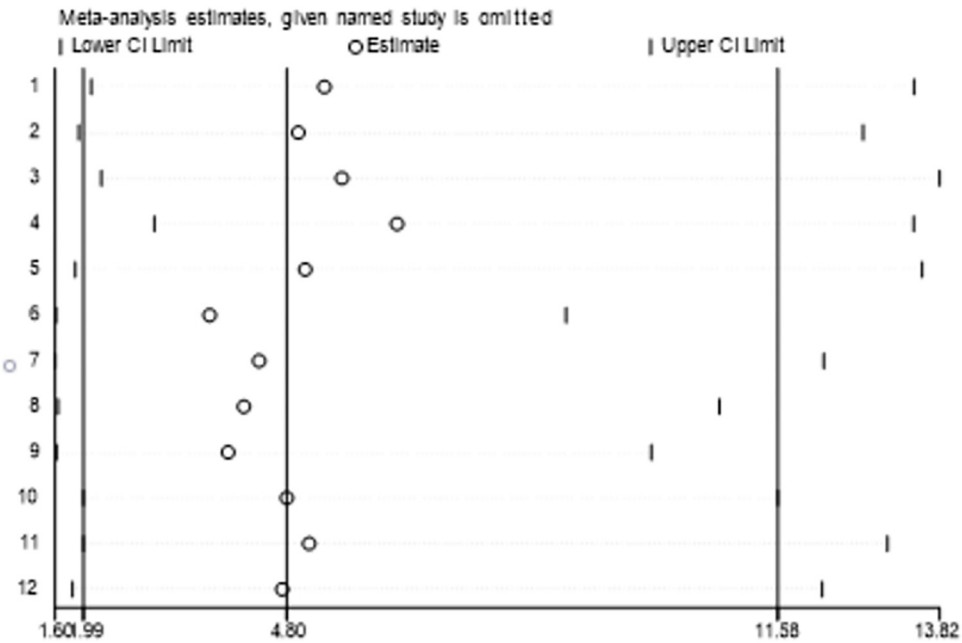

**Fig 4. Sensitivity analysis.**

However, the reported results are still controversial, especially in the recurrence and metastasis of CRC after surgery. Some studies believed that the presence of CTCs 24 hours and 7 days after surgery is not correlated with the recurrence and metastasis of CRC after surgery [19, 25], the presence of CTCs in the first year after surgery had no effect on disease progression [37]. Therefore, relevant literature was included in this study for meta analysis. Due to large heterogeneity, random effect model was adopted to merge data. The result shows that sensitivity pooled into 0.67[95%$CI$(0.62, 0.71)](Fig 4A), specificity pooled into 0.71[95%$CI$ (0.67, 0.72)](Fig 4B), DOR pooled into 4.97[95%$CI$ (2.11, 11.72)] (Fig 3C), AUC was 0.7395 (Fig 3A). These results suggest that peripheral blood CTCs have moderate monitoring value for postoperative recurrence and metastasis of CRC.

Due to the large heterogeneity, we conducted a subgroup analysis of the detection methods. The results suggested that the Membrane-Arrays method may have higher monitoring value compared with RT-PCR. However, the accuracy of this result was open to question. On the one hand, it may be the reason for the machine detection, on the other hand, may be the problem of the technician's operation that caused this result. A meta-analysis of the prognostic role of CTCs detected by RT-PCR in non-stage IV colorectal cancer had been done in the literature, showing that CTCs detected by RT-PCT had prognostic value for colorectal cancer [38]. Therefore, it cannot be directly explained that the CTC detected by Membrane-Arrays is more valuable for monitoring recurrence and metastasis.

CTC detection time point is also very important [39], but so far, the best detection time of CTCs for recurrence and metastasis after CRC is still unclear. As shown in a previous article, CTCs detected 7 days after surgery were closely associated with recurrence and metastasis [24]. Considering the number of included studies and clinical reality, we divided within one week after the operation and one week after the operation into two time points for subgroup analysis, and found that the specificity, DOR and area under the SROC curve in one week after the operation were all higher than those of the operation high within one week (Table 2). It is suggested that the CTCs detected one week after operation may be more related to recurrence

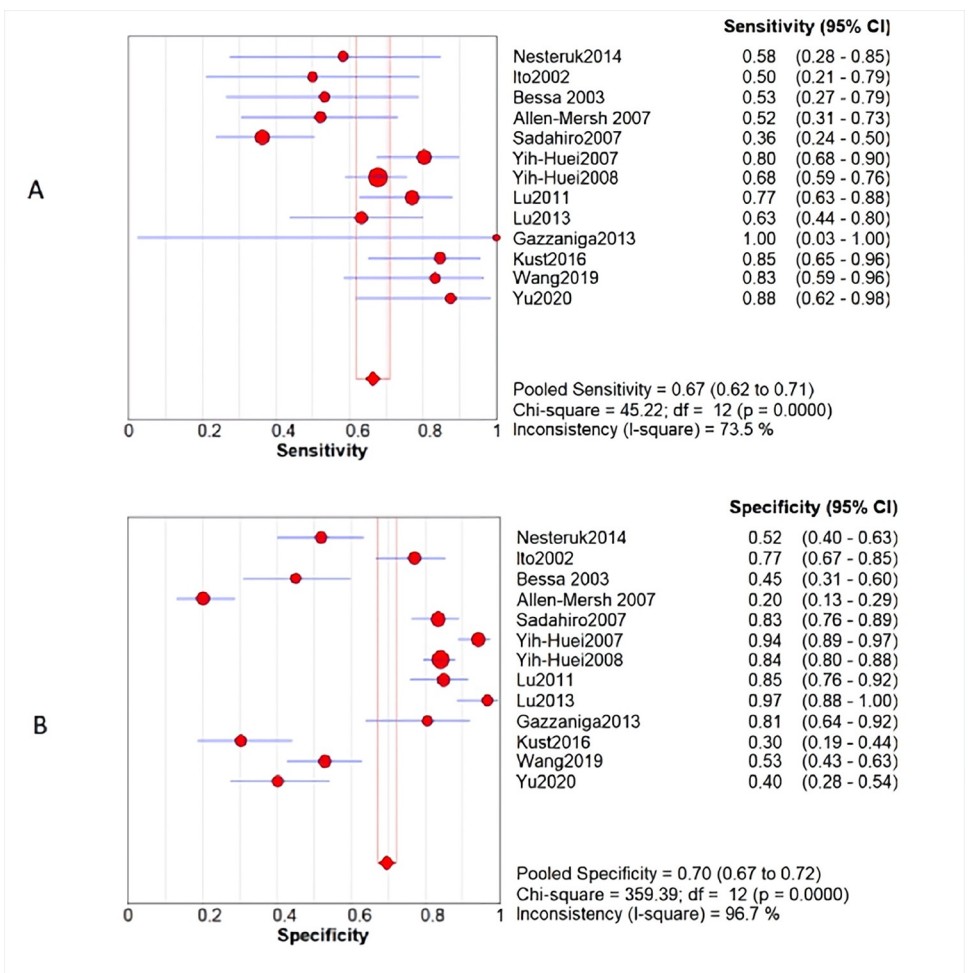

**Fig 5.** A: sensitivity B: specificity.

and metastasis. But CTC obtained one week post-surgery does not represent hematogenous spread during the procedure itself. More research in this area can be done in the future.

Generally speaking, +LR combination > 10 and -LR combination < 0.1 can basically confirm or rule out the diagnosis. In the results of this study, PLR was combined into 2.37 [95%CI (1.52, 3.71)] (Fig 6A), and NLR was combined into 0.53 [95%CI (0.36, 0.78)] (Fig 6B). It was suggested that positive CTCs cannot confirm the recurrence or metastasis of CRC, and negative CTCs cannot rule out the recurrence or metastasis of CRC.

As far as we know, this article is the first meta-analysis that directly analyzes the CTC to monitor the recurrence and metastasis of colorectal cancer after surgery. However, our results suggest that in non-stage IV CRC, its effectiveness in monitoring recurrence and metastasis can only be calculated medium (as Figs 3 and 4). Therefore, in clinical practice, the specific location of postoperative recurrence or metastasis of CRC still needs to be used according to the specific clinical practice. In addition, there was a significant correlation between the dynamic changes of CTCs detection and imaging results [40], it was better than imaging examination in terms of timeliness, especially in metastatic CRC, CTCs detection could find clues of tumor metastasis earlier than imaging examination [41]. Moreover, CTC combined with CEA, CA125, CA19-9 and other tumor markers have higher sensitivity and specificity, and may

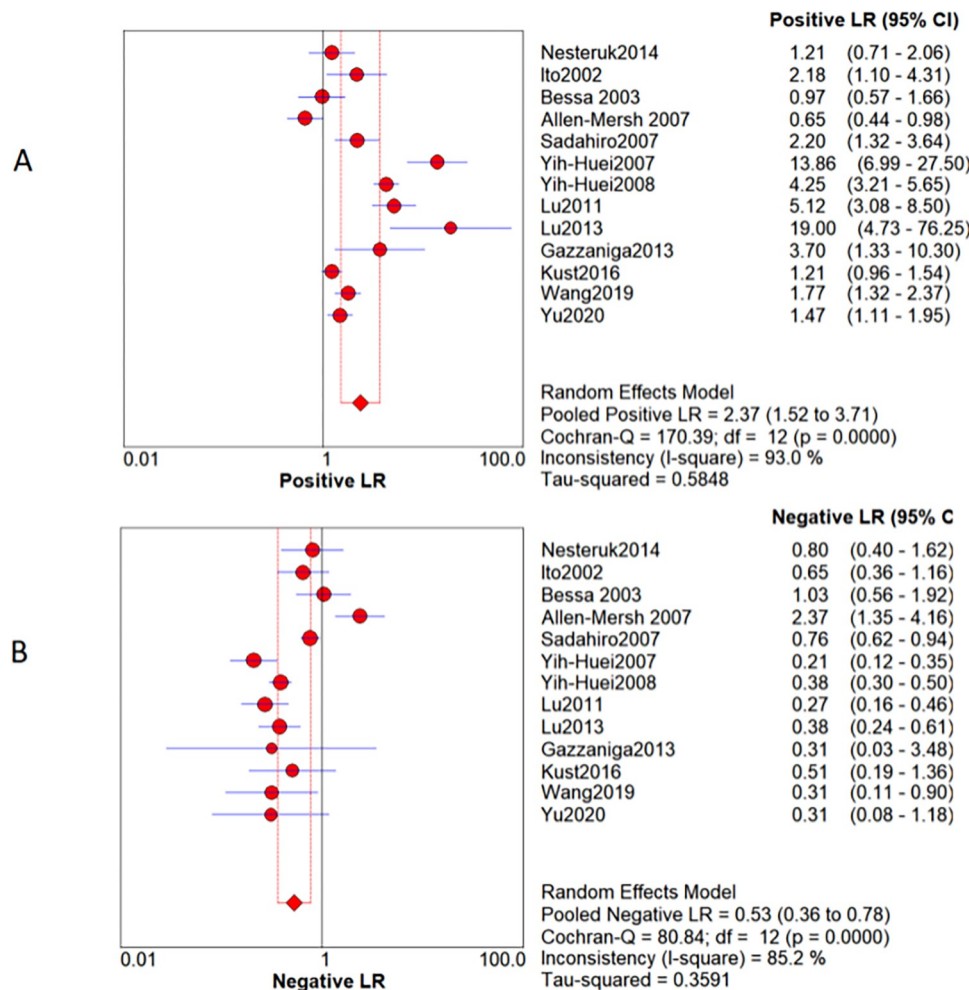

**Fig 6.** A: Positive likelihood ratio forest plot. B: Negative likelihood ratio forest plot.

**Table 2. Subgroup analysis.**

| Subgroups | article number | Pooled sensitivity (95% CI) | I²(%) sensitivity | Pooled specificity (95% CI) | I²(%) specificity | Pooled PLR(95% CI) | I²(%) PLR | Pooled NLR(95% CI) | ²(%) NLR | Pooled DOR | I²(%) DOR | AUC |
|---|---|---|---|---|---|---|---|---|---|---|---|---|
| CTCs detection method | | | | | | | | | | | | |
| Membrane-Arrays | 4 | 0.72(0.66–0.77) | 39.1 | 0.88(0.85–0.90) | 82.0 | 7.33(3.90–13.80) | 78.7 | 0.32(0.24–0.42) | 43.3 | 25.74 (10.38–63.88) | 76 | 0.8450 |
| RT-PCR | 6 | 0.52(0.44–0.61) | 72.2 | 0.55(0.51–0.59) | 96.6 | 1.24(0.88–1.75) | 71.8 | 0.92(0.62–1.37) | 69 | 1.42(0.62–3.65) | 73.2 | 0.5635 |
| Collection time | | | | | | | | | | | | |
| Within one week after surgery | 8 | 0.63(0.52–0.73) | 47.6 | 0.47(0.42–0.52) | 94.6 | 1.24(0.85–1.80) | 71.1 | 0.92(0.54–1.57) | 66.0 | 1.38(0.54–3.50) | 67.6 | 0.5868 |
| One week after surgery | 5 | 0.66(0.60–0.71) | 86.1 | 0.87(0.84–0.89) | 78.4 | 5.68(3.13–10.31) | 83.1 | 0.37(0.22–0.63) | 90.8 | 16.60(5.9–46.71) | 87.5 | 0.8778 |

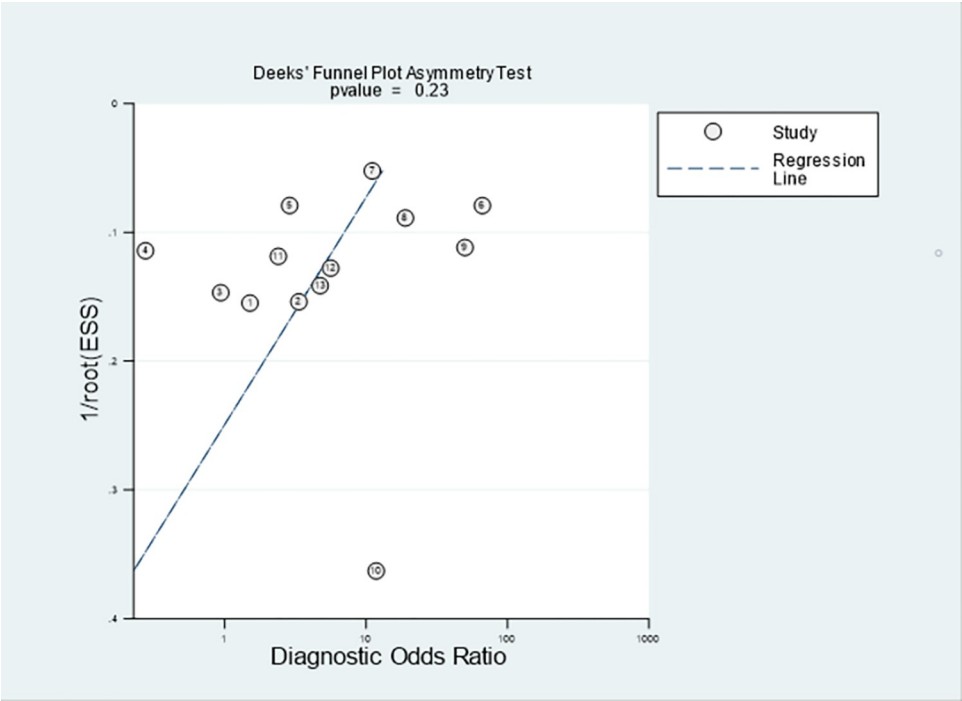

**Fig 7. Publication bias.**

have higher diagnostic value [42]. Therefore, CTCs cannot be used alone for the clinical diagnosis of CRC, nor can it replace other tumor markers and imaging examinations, but based on the results of CTCs detection, the frequency of these examinations can be reduced, or the time of these examinations can be prolonged, or combined with other tumor markers together with the detection to improve accuracy, the combination of CTCs detection and imaging can provide more clinical information [43]. It can be used as one of the simple, effective and non-invasive diagnostic methods.

However, there are still some limits in this article. First, only three databases were searched for literature search, and unpublished and other databases were not searched, which may lead to potential publication bias, although this article did not indicate publication bias. Second, the markers used to identify CTCs using these detection methods are different. The detection methods recognize CEA mRNA, CEA or CK20, CD45−, hTERT, CK-19, CK-20, CEA, etc. to identify CTCs. The diversity and differences in markers may be one of the reasons for heterogeneity. The sensitivity and specificity of each marker cannot be further analyzed in subgroups due to the fact that there is little relevant literature included. Third, local recurrence of colorectal cancer is still potentially curable while metastatic disease is not. However, only two of the included papers [21, 24] separated recurrence and metastasis, and the rest of the papers did not do so. Therefore, this study analyzed recurrence and metastasis together. Forth, the threshold of CTCs is different due to the difference in detection methods and markers. The inconsistency of the threshold may lead to large differences in results. This may also be the source of heterogeneity, and due to the comparison of related literature diffusion, so there is no further analysis, which is also one of the shortcomings of this research. Fifth, some studies believed that CTCs could be used as sensitive markers for adjuvant chemotherapy in patients with stage II CRCs [44], Gazzaniga et al. [28] also believed that CTCs can be used as markers for the selection of postoperative adjuvant chemotherapy for stage II-III CRC. Huang et al.

[45] performed a meta-analysis and found that, CTCs were related to the prognosis of CRC patients receiving chemotherapy, and CTCs could be used as predictive markers of chemotherapy response, that was, the disease control rate was significantly higher in CRC patients with CTC-low compared with CTC-high. However, in the literature included in this study, some studies do not clearly state whether post-operative adjuvant treatment is required. In addition, some studies only describe adjuvant treatment for high-risk stage II and stage III patients, but do not describe the treatment of patients in this category. During the follow-up period, the number of tumor recurrences and metastases, or the relationship between the number of recurrence and metastasis cases in these patients and CTCs, led to no further analysis of the literature on non-adjuvant and adjuvant treatment. Therefore, the results of this study are not yet available on whether CTCs can be used as response markers for post-operative chemotherapy; Hence, future research is required.

## Conclusion

Peripheral blood CTCs have a moderate monitoring value for recurrence and metastasis after CRC surgery; CTCs detected one week after surgery may be more correlated with recurrence and metastasis.

## Supporting information

**S1 Checklist. PRISMA 2020 checklist.**
(DOCX)

## Author Contributions

**Data curation:** Jiao Wu, Jianhua Zou, Liusheng Li.

**Formal analysis:** Kangjun Yi.

**Funding acquisition:** Yu Wu.

**Investigation:** Kangjun Yi, Jingyan Yang.

**Methodology:** Zhongyu Li.

**Resources:** Ning Cui.

**Software:** Tengteng Hao.

**Supervision:** Jianhua Zou, Liusheng Li, Yu Wu.

**Validation:** Ning Cui, Tengteng Hao, Yu Wu.

**Writing – original draft:** Jiao Wu, Zhongyu Li.

**Writing – review & editing:** Jiao Wu.

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
