## [Decision Letter · Decision Letter 0]

14 Apr 2022

PONE-D-21-38928A meta-analysis of the value of circulating tumor cells in monitoring postoperative recurrence and metastasis of colorectal cancerPLOS ONE

Dear Dr. WU,

Thank you for submitting your manuscript to PLOS ONE. After careful consideration, we feel that it has merit but does not fully meet PLOS ONE’s publication criteria as it currently stands. Therefore, we invite you to submit a revised version of the manuscript that addresses the points raised during the review process.

ACADEMIC EDITOR:  Please respond to the reviewers' comments. ==============================

We look forward to receiving your revised manuscript.

Kind regards,

Jason S. Gold

Academic Editor

PLOS ONE

“This study was supported by grants from the National key research and development plan of China (2017YFC1700606).”

“The Funders is the corresponding author of this manuscript and helped design it.”

“The Funders is the corresponding author of this manuscript and helped design it.”

5. PLOS requires an ORCID iD for the corresponding author in Editorial Manager on papers submitted after December 6th, 2016. Please ensure that you have an ORCID iD and that it is validated in Editorial Manager. To do this, go to ‘Update my Information’ (in the upper left-hand corner of the main menu), and click on the Fetch/Validate link next to the ORCID field. This will take you to the ORCID site and allow you to create a new iD or authenticate a pre-existing iD in Editorial Manager. Please see the following video for instructions on linking an ORCID iD to your Editorial Manager account: https://www.youtube.com/watch?v=_xcclfuvtxQ.

6. Please upload a copy of Figure2-7. If the figure is no longer to be included as part of the submission please remove all reference to it within the text.

Additional Editor Comments:

Please respond to the reviewers' comments.

Reviewers' comments:

Reviewer's Responses to Questions

**Comments to the Author**

1. Is the manuscript technically sound, and do the data support the conclusions?

Reviewer #1: Yes

Reviewer #2: Yes

2. Has the statistical analysis been performed appropriately and rigorously? 

Reviewer #1: Yes

Reviewer #2: Yes

3. Have the authors made all data underlying the findings in their manuscript fully available?

Reviewer #1: Yes

Reviewer #2: Yes

4. Is the manuscript presented in an intelligible fashion and written in standard English?

Reviewer #1: Yes

Reviewer #2: No

5. Review Comments to the Author

Reviewer #1: This meta-analysis by Wu et al examines the utility of CTC to predict recurrence or metastases of stage I-III CRC following surgery. Thirteen studies with 1788 patients were included in the analysis. CTC detected by membrane arrays and at least one week post-operatively had the best sensitivity, pooled PLR, pooled NLR, and pooled DOR for predicting recurrence and metastases. The emerging role of liquid biopsies in oncology clinical trials and management of patients makes the topic clinically relevant.

Major Comments:

1) Why was disease recurrence and metastases combined? Was this because of how the 13 included studies were designed? Since local disease recurrence is theoretically curable, while only a subset of metastatic disease is, it would be better if CTC predicting recurrence was separated from CTC predicting metastases.

2) Though I was able to find the PRISMA checklist, I did not find Figure 1: Search flowchart. Please include in revision.

3) Since there are multiple ongoing studies utilizing ctDNA to predict disease recurrence in resected CRC, it would benefit the discussion to compare/contrast CTC and ctDNA in this space.

4) How did the authors qualitatively conclude that peripheral blood CTC has a moderate monitoring value for predicting CRC recurrence and metastases following surgery? How would they incorporate CTC detection into their clinical practice?

Minor Comments:

1) What is meant by B-ultrasound?

2) Can the authors discuss why CTC were assayed around 7 days post-op? Theoretically, cancer cells embolized during surgery could still be present. In contrast, ctDNA is typically assayed 3-6 weeks post-op.

3) Some misspelled words, such as "literatures."

4) On pg 19, no mention of EpCAM as a marker of CTC was included in detection methods.

5) Please clarify the sentence on page 20, "a low number of CTCs often had a good response rate." Low CTC could represent lower stage disease pre-treatment, a less aggressive biology, etc. and not only a better treatment response.

Reviewer #2: Thank you for your work, yet there are some flaws to be addressed:

- the paper needs to be revised by a native English speaker

- Figure 1 is totally blurred and therefore it should be improved

- In the paper there are several misprints, please improve the manuscript

- the authors did not succeed in include a clinical impact of this research. Have you considered the implementation of molecular tumor board in resected setting? Please read this paper that could help you. PMID: 34896250

6. PLOS authors have the option to publish the peer review history of their article (what does this mean?). If published, this will include your full peer review and any attached files.

Reviewer #1: No

Reviewer #2: No

---

## [Author Response · Author response to Decision Letter 0]

2 Jun 2022

First of all, I would like to thank the editors and reviewers for taking the time to read my article and give professional opinions, and I will answer the questions raised below.

Major Comments:

1)In fact, I agree with the opinion of reviewer 1, and I wanted to do so at first. However, only two of the included papers, Nesteruk2014 and Yih-Huei2007, separate recurrence and metastasis, and the rest of the papers did not do so. Therefore, this study analyzed recurrence and metastasis together.

2)The flow chart has been uploaded again in the article.

3)I am very grateful for the opinions of the reviewers. At that time, the monitoring of ctDNA for postoperative recurrence and metastasis of colorectal cancer was not carefully found. In this study, the literature was reviewed to supplement the article.

4)The reason why the moderate monitoring value of CTC is obtained is mainly based on the AUC. The diagnostic value is moderate: AUC = 0.7～0.9. Our results suggest that in non-stage IV CRC, effectiveness of CTC in monitoring recurrence and metastasis can only be calculated medium. Therefore, in clinical practice, the specific location of postoperative recurrence or metastasis of CRC still needs to be used according to the specific clinical practice. In addition, there was a significant correlation between the dynamic changes of CTCs detection and imaging results.it was better than imaging examination in terms of timeliness, especially in metastatic CRC, CTCs detection could find clues of tumor metastasis earlier than imaging examination. Moreover, CTC combined with CEA, CA125, CA19-9 and other tumor markers have higher sensitivity and specificity, and may have higher diagnostic value. Therefore, CTCs cannot be used alone for the clinical diagnosis of CRC, nor can it replace other tumor markers and imaging examinations, but based on the results of CTCs detection, the frequency of these examinations can be reduced, or the time of these examinations can be prolonged, or combined with other tumor markers together with the detection to improve accuracy, the combination of CTCs detection and imaging can provide more clinical information. It can be used as one of the simple, effective and non-invasive diagnostic methods.

Minor Comments:

1)I'm sorry that B-ultrasound didn't make it clear, in fact it refers to Ultrasonography, the article has been revised. 

2)CTC detection time point is also very important, but so far, the best detection time of CTCs for recurrence and metastasis after CRC is still unclear. Considering the number of included studies and clinical reality, we divided within one week after the operation and one week after the operation into two time points for subgroup analysis.

3)Sorry, there will be some grammatical errors in it as my native language is not English, I have fixed it.

4)In the included literature, there is no article that clearly suggests that EpCAM is used as a marker, so this article does not mention it.

5)Yes, the reviewer's opinion is correct, I have revised it.

---

## [Decision Letter · Decision Letter 1]

19 Jul 2022

PONE-D-21-38928R1A meta-analysis of the value of circulating tumor cells in monitoring postoperative recurrence and metastasis of colorectal cancerPLOS ONE

Dear Dr. WU,

Thank you for submitting your manuscript to PLOS ONE. After careful consideration, we feel that it has merit but does not fully meet PLOS ONE’s publication criteria as it currently stands. Therefore, we invite you to submit a revised version of the manuscript that addresses the points raised during the review process.

ACADEMIC EDITOR: Please address reviewer's comments. In particular, please edit the written English.

We look forward to receiving your revised manuscript.

Kind regards,

Jason S. Gold

Academic Editor

PLOS ONE

Journal Requirements:

Additional Editor Comments:

Please address the reviewer's comments. In particular, please edit the written English.

Reviewers' comments:

Reviewer's Responses to Questions

**Comments to the Author**

1. If the authors have adequately addressed your comments raised in a previous round of review and you feel that this manuscript is now acceptable for publication, you may indicate that here to bypass the “Comments to the Author” section, enter your conflict of interest statement in the “Confidential to Editor” section, and submit your "Accept" recommendation.

Reviewer #1: (No Response)

Reviewer #2: (No Response)

2. Is the manuscript technically sound, and do the data support the conclusions?

Reviewer #1: Yes

Reviewer #2: Yes

3. Has the statistical analysis been performed appropriately and rigorously? 

Reviewer #1: Yes

Reviewer #2: Yes

4. Have the authors made all data underlying the findings in their manuscript fully available?

Reviewer #1: Yes

Reviewer #2: Yes

5. Is the manuscript presented in an intelligible fashion and written in standard English?

Reviewer #1: Yes

Reviewer #2: Yes

6. Review Comments to the Author

Reviewer #1: The authors have addressed many of my prior critiques. Please revise by incorporating responses to the comments below.

Major comments:

1) Please include your response to my prior critique about separating recurrence and metastases in the discussion section. This point is important for readers to understand. Locally recurrent disease is still potentially curable while metastatic disease is not.

2) I appreciate the added comments regarding ctDNA. The most recent NCCN guidelines include this discussion regarding ctDNA

A prospective, multicenter study of 130 patients with stage I–III colon cancer detected ctDNA by multiplex, PCR-based next-generation sequencing (NGS).306 Thirty days after surgery, patients with positive ctDNA assays were seven times morelikely to experience disease relapse than patients who were ctDNAnegative (HR, 7.2; 95% CI, 2.7–19.0; P < .001). Likewise, after adjuvant chemotherapy, patients with ctDNA-positive assays were 17 times more likely to have disease relapse (HR, 17.5; 95% CI, 5.4–56.5; P < .001). Another prospective study of 150 patients with localized colon cancerdetected dtDNA with NGS following surgery.307 In this study, detection of ctDNA was also associated with poorer DFS (HR, 17.56).

It would be helpful to include these studies in the discussion and more extensively discuss the advantages/disadvantages of ctDNA vs CTC as ctDNA is being rapidly incorporated into clinical trials while CTC is less so. The point of this meta-analysis is to convince readers that CTC should also be considered as a rational and practical liquid biopsy.

3) The response to my prior critique that "CTC combined with CEA, CA125, CA19-9 and other tumor markers has higher sensitivity and specificity" should be included in the discussion. Again, to emphasize that there are alternatives to ctDNA.

4) Please include a statement somewhere in the discussion that shows that CTC obtained one week post-surgery does not represent hematogenous spread during the procedure itself. I'm still skeptical that one week post-op is the best time to sample CTC which represent metastatic potential.

Minor comment:

1) The manuscript needs further editing of written English. For example in the first paragraph of the discussion, I believe "pathogenesis" is meant when "pathogens" is used. Another example is on page 18 where "litter" precedes "relevant literature." I do not understand that meaning.

Reviewer #2: thank you for your valuable work.

The authors have addressed all the reviewers' comments.

The paper is well-written and interesting to the general audience.

7. PLOS authors have the option to publish the peer review history of their article (what does this mean?). If published, this will include your full peer review and any attached files.

Reviewer #1: No

Reviewer #2: No

---

## [Author Response · Author response to Decision Letter 1]

27 Jul 2022

First of all, I would like to thank the editors and reviewers for taking the time to read my article and provide valuable comments, from which I have benefited a lot. Next, please allow me to answer the questions of the editors and reviewers.

(1)I have checked the references in the article, there should be no retractions, I cite the literature strictly according to the requirements of your journal.

(2)Regarding the opinions of the reviewer, I have added them in the discussion part of the article. The red part of the article is written according to the opinions of the reviewer.

---

## [Decision Letter · Decision Letter 2]

25 Aug 2022

A meta-analysis of the value of circulating tumor cells in monitoring postoperative recurrence and metastasis of colorectal cancer

PONE-D-21-38928R2

Dear Dr. WU,

We’re pleased to inform you that your manuscript has been judged scientifically suitable for publication and will be formally accepted for publication once it meets all outstanding technical requirements.

Kind regards,

Jason S. Gold

Academic Editor

PLOS ONE

Additional Editor Comments (optional):

The authors have appropriately responded to all the reviewers' questions and comments.

Reviewers' comments:

Reviewer's Responses to Questions

**Comments to the Author**

1. If the authors have adequately addressed your comments raised in a previous round of review and you feel that this manuscript is now acceptable for publication, you may indicate that here to bypass the “Comments to the Author” section, enter your conflict of interest statement in the “Confidential to Editor” section, and submit your "Accept" recommendation.

Reviewer #1: All comments have been addressed

2. Is the manuscript technically sound, and do the data support the conclusions?

Reviewer #1: Yes

3. Has the statistical analysis been performed appropriately and rigorously? 

Reviewer #1: Yes

4. Have the authors made all data underlying the findings in their manuscript fully available?

Reviewer #1: Yes

5. Is the manuscript presented in an intelligible fashion and written in standard English?

Reviewer #1: Yes

6. Review Comments to the Author

Reviewer #1: Prior critiques have been addressed appropriately by the authors. After editorial review for grammar and spelling, would be appropriate for publication.

7. PLOS authors have the option to publish the peer review history of their article (what does this mean?). If published, this will include your full peer review and any attached files.

Reviewer #1: No

---

## [Editor Report · Acceptance letter]

6 Sep 2022

PONE-D-21-38928R2 

A meta-analysis of the value of circulating tumor cells in monitoring postoperative recurrence and metastasis of colorectal cancer 

Dear Dr. WU:

I'm pleased to inform you that your manuscript has been deemed suitable for publication in PLOS ONE. Congratulations! Your manuscript is now with our production department. 

Kind regards, 

on behalf of

Dr. Jason S. Gold 

Academic Editor

PLOS ONE